# High Rate of Human T-Cell Lymphotropic Virus-2 in Patients with HIV in the Peruvian Amazon

**DOI:** 10.3390/tropicalmed10090267

**Published:** 2025-09-17

**Authors:** Silvia Otero-Rodriguez, Martin Casapia-Morales, Carmen de Mendoza, Viviana Pinedo-Cancino, Seyer Mego-Campos, Vicente Soriano, Esperanza Merino, José-Manuel Ramos-Rincón

**Affiliations:** 1Infectious Diseases Unit, Doctor Balmis University General Hospital, 03010 Alicante, Spain; merino_luc@gva.es (E.M.); jose.ramosr@umh.es (J.-M.R.-R.); 2Institute of Sanitary and Biomedical Research (ISABIAL), 03010 Alicante, Spain; 3Infectious Diseases and Tropical Medicine Service, Loreto Regional Hospital, Iquitos 16001, Peru; mcasapia@acsaperu.org; 4Medical Department, Asociación Civil Selva Amazónica, Iquitos 16001, Peru; 5Faculty of Human Medicine, National University of the Peruvian Amazon, Iquitos 496, Peru; viviana.pinedo@unapiquitos.edu.pe; 6Puerta de Hierro University Hospital & Research Foundation-IDIPHISA, 28222 Madrid, Spain; cmendoza.cdm@gmail.com; 7Laboratory for Research on Natural Antiparasitic Products of the Amazon (LIPNAA-CIRNA), National University of the Peruvian Amazon, Iquitos 496, Peru; megoseyer@gmail.com; 8UNIR Health Sciences School and Medical Center, 28224 Madrid, Spain; vicente.soriano@unir.net; 9Clinical Medicine Department, Miguel Hernández University of Elche, 03202 Elche, Spain

**Keywords:** HTLV, HTLV-2, human T-cell lymphotropic virus, human immunodeficiency virus, HIV, Peru, Amazon

## Abstract

HTLV-1/2 in people with HIV (PWH) has been little studied in the Peruvian Amazon, an endemic area for both viruses. We aimed to estimate its prevalence and describe the main clinical and epidemiological features of individuals with HTLV-HIV co-existence. We conducted a cross-sectional study (October–December 2023) at the Division of Infectious Diseases and Tropical Medicine at the Regional Hospital of Loreto in Iquitos. We performed a screening test (recombinant HTLV I+II ELISA) and confirmed the results with INNO-LIA. Among 293 PWH analyzed, 14 (4.8%) were HTLV-positive: 1/293 was HTLV-1-positive (0.3%; 95% CI 0.06–0.9), 11/293 were HTLV-2-positive (3.8%; 95% CI 2.1–6.8), and 2/293 were non-typeable (0.7%; 95% CI 0.1–2.7). Compared with HIV-monoinfected individuals, superinfected patients were older (55 vs. 39 years; *p* = 0.001). Low education was more frequent in the univariate analysis (35.7% vs. 15.4%; *p* = 0.05) but was not retained in the multivariable model. In conclusion, HIV–HTLV-2 co-existence is relatively common (~4%) in the Peruvian Amazon, particularly among older individuals, highlighting the need for targeted screening and prevention strategies. Integrating HTLV testing into routine HIV clinic workflows, along with brief and focused counseling for superinfected patients, may help optimize follow-up and care.

## 1. Introduction

Infection with Human T-Cell Lymphotropic Virus Types I and II (HTLV-1/2) represents an underestimated public health issue, with a heterogeneous geographic distribution and a significant burden in endemic areas of Latin America, particularly in Brazil and Peru. HTLV-1 affects more than 10 million people worldwide [1] and is associated with severe conditions such as adult T-cell leukemia/lymphoma and HTLV-1–associated myelopathy. In contrast, HTLV-2, which is considered less pathogenic, predominates among Indigenous populations and people who inject drugs [2,3,4,5,6,7,8].

HTLV-1/2 is primarily transmitted through sexual contact, vertical transmission (particularly via breastfeeding), blood transfusions, and, in certain regions, through exposure to contaminated blood through unsterile practices, such as scarification rituals or poorly hygienic procedures. These transmission routes are shared with the human immunodeficiency virus (HIV) [2].

The presence of both HIV and HTLV-1 in an individual’s immune system promotes immune dysregulation, characterized by paradoxical expansion of dysfunctional CD4^+^ and CD8^+^ T cells and chronic immune activation, ultimately leading to immune exhaustion despite elevated lymphocyte counts [9]. While some reports suggest that HIV–HTLV-2 coinfection may be linked to a ‘long-term non-progressor’ phenotype, evidence remains inconclusive and clinically controversial [1]. These interactions may influence HIV progression and outcomes, underscoring the importance of HTLV testing in HIV-positive patients—particularly those with compatible ethnic backgrounds and unexpectedly high CD4 counts—and the need for vigilance regarding HTLV-associated conditions such as HAM/TSP, adult T-cell leukemia/lymphoma, and *Strongyloides* hyperinfection [10,11].

In Brazil, the prevalence of HTLV/HIV co-existence is heterogeneous, ranging from 1.3% to 7%. HTLV-1a is the predominant subtype in the Amazon region, while a high endemicity of HTLV-2 has been observed among Indigenous populations and drug users. In Peru, although data are more limited, HTLV-1/2 infection has been documented in both urban and rural populations, with transmission hotspots in the Amazon region and evidence of circulation among Indigenous communities and individuals with similar risk factors to those reported in Brazil [5,6,7,10,12,13,14].

The Peruvian Amazon is a key area for HIV transmission, with the second-highest cohort of patients in the country receiving antiretroviral treatment after Lima [15]. Some studies have also revealed very high HIV rates in isolated indigenous populations, mainly due to sexual exposure without protection (including polygamy and homosexual practices), as there is little knowledge about the infection [16,17]. The absence of systematic screening programs for HIV and limited clinical awareness hinder detection and control of the infection, highlighting the urgent need to strengthen epidemiological surveillance and research efforts in the region [2,3,4,5].

The aim of this study is to assess the prevalence of HTLV-1/2 infection in a pilot sample of PWH in the Peruvian Amazon, describe the clinical and epidemiological characteristics of superinfected individuals, and analyze differences compared to HIV-monoinfected patients.

## 2. Materials and Methods

### 2.1. Study Design and Setting

A cross-sectional study was conducted among PWH who were receiving care at the Division of Infectious Diseases and Tropical Medicine of the Regional Hospital of Loreto “Felipe Santiago Arriola Iglesias” in Iquitos, Peru. The study period was from 20 October 2023 to 31 December 2023.

### 2.2. Study Population and Enrollment Criteria

Adults aged 18 years and older with confirmed HIV infection who were receiving outpatient care at the Regional Hospital of Loreto were eligible for inclusion. After providing informed consent, participants completed a semi-structured oral interview that collected data on demographics, clinical history and potential epidemiological risk factors. After that, a blood sample was obtained.

We planned to estimate the prevalence of HTLV coinfection among ~1000 PWH in care. Using an expected prevalence of 10% according to a previous meta-analysis in Peru [18], 95% confidence, and ±3% absolute precision, the required sample for a single proportion was 384. After applying the finite-population correction for N = 1000, this became 278. Allowing 6% for non-evaluable/losses, the final target sample was ≈295 participants.

Separated serum samples were aliquoted and frozen at −20 °C. When 96 samples were obtained, an ELISA kit was used, never exceeding four weeks. The presence of antibodies against HTLV-1/2 was initially screened using a single ELISA assay (HTLV I+II ELISA recombinant v.4.0 96-well kit, Wiener Lab, Rosario, Argentina; catalog number 1671096). A result was considered positive when the optical density value exceeded the negative control by 0.200, according to the manufacturer’s instructions. Due to the lack of local confirmatory testing, the 15 serum samples that were positive for HTLV at screening were sent to a HTLV reference laboratory in Puerta de Hierro University Hospital, Madrid, Spain. To facilitate safe transport from Iquitos without a cold chain, aliquots of frozen serum were applied to Whatman filter paper to create dried serum spots (DSS), which were air-dried, sealed with desiccant, and shipped using standard triple packaging. Upon arrival, DSS were eluted and tested with INNO-LIA HTLV I/II Score (Fujirebio, Tokyo, Japan; catalog number 80540), following standard laboratory procedures. While serum/plasma are the manufacturer’s validated matrices, DSS were used exclusively for transport. Molecular assays, including HTLV typing and proviral load, could not be performed, as DNA extracted from cellular fractions (whole blood/PBMCs) was not collected.

We performed serologic testing for *Strongyloides stercoralis* because of its epidemiologic overlap with HTLV in the Amazon and evidence that HTLV-1 coinfection increases the risk of severe/disseminated strongyloidiasis and treatment failure [19,20]. Accordingly, we assessed *Strongyloides*–HTLV coinfection among PWH to inform clinical follow-up in endemic settings. Serology for *S. stercoralis* was performed using a commercial IgG ELISA (Strongyloides IgG IVD ELISA, DRG Instruments GmbH, Marburg, Germany; catalog number EIA-4208), following the manufacturer’s instructions; results were interpreted using kit-specified cutoffs.

Those who were positive for HTLV 1/2 were contacted again to conduct a more comprehensive interview on risk factors for transmission, including the origin of their parents.

### 2.3. Statistical Data Analysis

Categorical variables were summarized as frequencies and percentages, while continuous variables were expressed as medians and interquartile ranges (IQRs). Ninety-five percent confidence intervals (95% CIs) were calculated using the Newcombe method. Comparisons between categorical variables were performed using the Chi-square test or Fisher’s exact test when any expected cell count was <5, while continuous variables were analyzed using the Mann–Whitney U test, given the non-normal distribution. Age was analyzed as a continuous variable and, for specific comparisons, dichotomized at the 75th percentile (P75 = 49 years) into <50 vs. ≥50 years. Risk factors associated with HTLV positivity were explored using bivariate analysis, with odds ratios (ORs) used to quantify associations. Subsequently, we fitted a multivariable logistic regression model using a forward stepwise procedure to identify independent risk factors for HTLV positivity. Age and sex were included a priori, and additional covariates with *p* < 0.10 in the univariable analyses were considered eligible for inclusion in the model. Statistical analyses were performed via IBM SPSS Statistics, version 22.0 (IBM Corp., Armonk, NY, USA).

### 2.4. Ethical Considerations

The study protocol was approved by the Ethics Committee of the Regional Hospital of Loreto in Iquitos, Peru (EXP: ID-018-CIEI-2023) and by the Ethics and Research Integrity Committee of Miguel Hernández University of Elche, Spain (DMC.JMRR.230908). Written informed consent was obtained from all participants. Confidentiality of data was strictly maintained, and results were only disclosed to each participant’s HIV care provider, who ensured appropriate follow-up and treatment.

## 3. Results

### 3.1. Overview of the Study Population

A total of 293 PWH were included in the study, of whom 66.9% were male, with a median age of 40 years (IQR 30–49). Of the participants, 16.7% had no formal education or had only attended primary school, 21.8% had received a blood transfusion, 94.5% had been breastfed, and 89.9% acquired HIV through sexual transmission. This overview serves as a reference for subsequent subgroup comparisons between HTLV-positive and HTLV-negative participants, described in Table 1.

### 3.2. HTLV Subtypes

15/293 patients tested positive for HTLV during screening. Of them, 14/293 (4.8%) were definitively confirmed: 1/293 was HTLV-1 (0.3% 95% CI 0.06–0.9), 11/293 were HTLV-2 (3.8%, 95% CI 2.1–6.8) and 2/293 (0.7%, 95% CI 0.1–2.7) were non-typeable by INNO-LIA HTLV I/II score (Figure 1).

### 3.3. Description of HTLV-HIV Co-Existence

Of the 14 patients confirmed positive for HTLV, the median age was 55 years (IQR 52–61), and >85% were older than 50 years. All but one were Mestizos (92.9%), while one patient was of the Kukuma race, a tribe that lives on the Marañón River inside Pacaya Samiria National Reserve (Figure 2). The parents of 6 of 12 patients with information available (50%) came from the tributaries of the Amazon River, in southern Iquitos, while the origin of one family was Pebas, in the part of the Amazon River that heads towards the border with Brazil, in northern Iquitos. 4 of the patients had parents from Iquitos, and 2 had no recorded data on their origin due to loss to follow-up. These findings may indicate that HTLV is not confined to river basins bordering Brazil, but circulating along the Marañón and Ucayali Rivers, involving both native communities and mestizo populations in Iquitos with family ties to endemic riverside areas.

All but 1 (92.9%) had been breastfed; 3 (21.8%) had received a blood transfusion, and only 1 (7.1%) referred to scarification practices. Four (28.6%) patients had non-heterosexual sexual practices or more than five sexual partners. The characteristics of HIV-HTLV co-existence are presented in Table 2.

### 3.4. HTLV-Positive vs. HTLV-Negative Patients

Table 1 presents the differences between PWH with positive and negative HTLV-1/2 serology. In the univariate analysis, the risk factors associated with HTLV-1/2 positivity in the screening were age ≥ 50 years (85.7% vs. 21.5%) (*p* = 0.001) (OR 21.9; 95% CI 4.77–100) and lack of formal education or only graduated from primary school (35.7% vs. 15.4%) (*p* = 0.05) (OR 2.96 95% CI 0.95–9.27). In the multivariable logistic regression—adjusting for sex and all covariates with *p* < 0.10 in univariable screening—age was the only independent predictor. Using a dichotomous cut-off, age ≥ 50 years was associated with the outcome (aOR 22.29; 95% CI 4.66–106). Modeled as a continuous variable, each additional year increased the odds by 1.1% (aOR 1.11; 95% CI 1.05–1.17) (Figure 2).

**Table 2 tropicalmed-10-00267-t002:** Characteristics of patients with HIV-HTLV co-existence.

N	Type of HTLV	Age	Sex	Ethnicity	Origin of Parents	Breast-Feeding	Sexual Behavior/ Number of Sexual Partners	Non-Sterilized Procedures ^a^	Transfusion	Living in Rural Area ^b^	Chronic Hepatitis	STI ^c^	CD4 Count Nadir/Last	HIV Viral Load
1	HTLV-1	62	M	Mestizo	Tarapoto	Yes	Transgender/<5	No	Yes	No	No	No	479/677	<20
2	HTLV-2	52	M	Mestizo	Iquitos	Yes	Homosexual/<5	No	No	No	No	No	218/674	<20
3	HTLV-2	56	M	Kukuma	Marañón River	Yes	Homosexual/<5	No	No	No	No	No	287/684	<20
4	HTLV-2	60	F	Mestizo	Nauta	Yes	Heterosexual/≥5	No	No	No	No	No	113/113	<20
5	HTLV-2	61	M	Mestizo	Requena	Yes	Heterosexual/≥5	No	Yes	No	No	No	134/322	<20
6	HTLV-2	53	M	Mestizo	Cuzco	Yes	Bisexual/≥5	No	Yes	No	Yes	Gonorrhea Syphilis	52/371	<20
7	HTLV-2	60	M	Mestizo	LOF	Yes	Heterosexual/LOF	LOF	No	No	Yes	No	NA	<20
8	Non-typable HTLV	43	F	Mestizo	Pebas	Yes	Heterosexual/<5	No	No	No	No	No	261/261	<20
9	Non-typable HTLV	55	M	Mestizo	Iquitos	Yes	Heterosexual/<5	No	No	No	No	Gonorrhea Syphilis	455/455	<20
10	HTLV-2	54	F	Mestizo	Marañón River	Yes	Heterosexual/≥5	Scarification	No	No	No	Syphilis	76/525	<20
11	HTLV-2	64	M	Mestizo	LOF	Yes	Heterosexual/LOF	LOF	No	No	No	No	171/399	<20
12	HTLV-2	66	F	Mestizo	Ucayali River	Yes	Heterosexual/<5	No	No	No	No	No	519/519	<20
13	HTLV-2	45	M	Mestizo	Iquitos	Yes	Heterosexual/<5	No	No	No	No	No	434/434	<20
14	HTLV-2	50	M	Mestizo	Iquitos	No	Heterosexual/<5	No	No	No	No	Gonorrhea	344/344	<20

^a^ Non-sterilized procedures: injection, scarification, tattoos, dental procedures, intravenous drugs. ^b^ Defined as the absence of paved streets. ^c^ Sexually transmitted infections. LOF: not available due to loss to follow-up. NA: not available in the clinical history.

## 4. Discussion

This study confirms the relevance of HTLV-HIV coexistence in patients from the Peruvian Amazon, where HTLV-2 predominates over HTLV-1, in contrast to other regions of South America and the world where HTLV-1 is by far more frequent. Two distinct epidemiological patterns of HTLV-2/HIV superinfection have been described: one in Europe, largely associated with people who inject drugs [21], and another in Latin America, particularly in Brazil and Peru. In this region, the prevalence of HTLV-2 among people with HIV is variable and may exceed 3% in certain cohorts [2,8,9,11,12,13,14,22,23], with especially high rates reported in indigenous populations of the Amazon, although HIV/HTLV-2 cases have also been documented in urban settings.

In Peru, the Shipibo-Konibo ethnic group in the Amazonian region exhibits a high prevalence of HTLV-1 (5.7%) and HTLV-2 (3.8%) [24]. In other indigenous communities of the Peruvian Amazon, seroprevalence rates of 4.54% for HTLV-1 and 2.38% for HTLV-2 have been documented [25]. Similarly, cross-sectional studies in the Brazilian Amazon have shown a higher prevalence of HTLV-2 than HTLV-1, with seroprevalence rates ranging from 0% to 40% [8]. In general, the overall prevalence of HTLV-2 infection in the indigenous communities of the Amazonian Brazil ranges from 5.7% [26] to 8.1% [6]. Abreu et al. reported a prevalence for HTLV-2 of 18.5% and HTLV-1 of 0.13% in 1452 individuals from the Kayapó ethnic group, and found evidence of intrafamilial transmission in 42.7% of cases [6]. The variability in reported prevalence rates in the literature reflects differences in diagnostic methods, inclusion criteria, and the representativeness of the studied populations. In the Peruvian Amazon, the high proportion of indeterminate INNO-LIA HTLV I/II results underscores the need for confirmatory molecular testing and cautious interpretation of serological results, in line with the recommendations of the Infectious Diseases Society of America and the American Society for Microbiology [25,26]. Additionally, underreporting and the lack of systematic screening hinder the precise estimation of disease burden and the identification of emerging risk factors [8,23,27].

The older age observed in patients with HTLV-HIV co-existence has been observed in previous studies [28]. This may be due to the transmission pattern of HTLV-2, which may be favored by certain practices that were more frequently practiced in indigenous or marginal urban communities some decades ago [29]. In addition, HTLV-2 has lower rates of sexual transmission than HTLV-1 and HIV, with a lower potential for spreading in highly mobile populations with risky sexual behavior, but greater for spreading in those with longer periods of exposure to cumulative risk factors [28]. Although the best-documented risk factors for HTLV-2 infection in Amazonian indigenous populations are age and intravenous drug use, some cultural and socioeconomic factors have been previously associated with HTLV-1 infection. In a cohort of Peruvian women, low educational level (primary education or less) was significantly associated with HTLV-1 infection [30]. In a 10-year analysis from Brazil, an increasing trend in HTLV-1/2 seropositivity was associated with the lowest educational level, which is consistent with our trend [31] and may reflect disparities in access to health information and preventive practices, influencing transmission dynamics [32].

Although at least half of the patients’ relatives resided in rural communities along the tributaries and the main course of the Amazon River. The most frequent origin was rural settlements south of Iquitos along the Marañón and Ucayali Rivers, whereas communities along the Amazon River en route to Brazil were less common, which may support the existence of a persistent transmission niche in the Peruvian Amazon [2,5,25].

The clinical course of HTLV-2/HIV co-existence is heterogeneous. Although HTLV-2 is associated with lower pathogenicity than HTLV-1, some studies suggest that it may modulate HIV progression, with reports of “long-term non-progressor” phenotypes and lower HIV proviral load in individuals with HTLV-2, possibly mediated by increased CD8^+^ T cell cytotoxic activity. However, the evidence regarding the clinical impact of superinfection remains controversial and is limited by study heterogeneity and the lack of longitudinal follow-up [9,25].

One of the main strengths of this study is its focus on epidemiological surveillance strategies for HTLV and HIV, particularly in a region where such data are scarce. By addressing this neglected area, the study contributes valuable information for public health decision-making and future research planning in the Amazon region. However, the study has several limitations. First, the small sample size limits the generalizability of the findings. Second, confirmatory testing was performed on eluates from dried serum spots applied to Whatman paper, a transport method not formally validated by the INNO-LIA manufacturer and which may slightly reduce sensitivity in low-titer samples. However, previous studies using dried-spot matrices for HTLV serology report high specificity and variable sensitivity (≈81–100%), supporting their feasibility in field settings. The results should therefore be interpreted cautiously and confirmed with serum or plasma when possible [33]. Another limitation is the inability to perform molecular characterization (HTLV-1/2 typing by PCR and proviral-load quantification), because cellular specimens (whole blood/PBMCs) were not collected. Reliance on serology alone may leave a small proportion of cases untyped and precludes analyses relating proviral burden to clinical outcomes. Additionally, given the cross-sectional design, it was not possible to assess the longitudinal clinical impact, particularly the role of HTLV in HIV disease progression and related complications.

In conclusion, our screening study in the Peruvian Amazon highlights a significant prevalence of HIV–HTLV co-existence, particularly with HTLV-2. Risk factors include age over 50 years. However, awareness and identification of HTLV co-existence remain important, as they can guide appropriate patient follow-up and care strategies. Future multicenter studies with larger cohorts and long-term follow-up are needed to better define the clinical relevance of both HTLV-1 and HTLV-2 in people living with HIV, especially in endemic areas such as the Amazon region. From a public-health perspective, the ~4% HIV–HTLV coinfection burden supports integrating HTLV testing into HIV clinic workflows, linking positive cases to counseling and *Strongyloides* assessment, and strengthening referral networks. Programs should prioritize older PWH in endemic regions, include partner testing, and capture cases in surveillance systems to guide resource allocation and cost-effective screening strategies.

## Figures and Tables

**Figure 1 tropicalmed-10-00267-f001:**
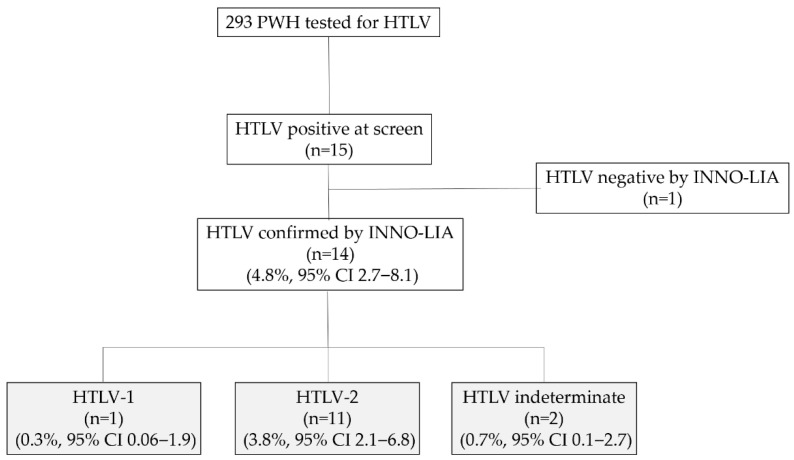
Flow chart of patients with HIV (PWH) who participated in the study and HTLV subtypes by INNO-LIA HTLV I/II score (number of cases, percentage and 95% confidence interval).

**Figure 2 tropicalmed-10-00267-f002:**
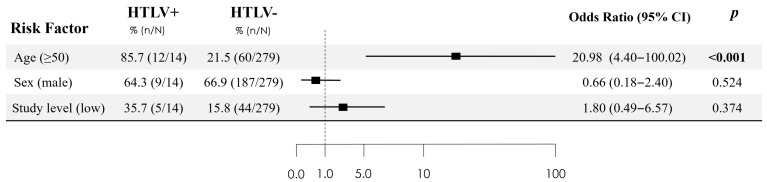
Independent predictors of HTLV infection from multivariable logistic regression analysis. Statistically significant differences (*p* value < 0.05) are shown in bold.

**Table 1 tropicalmed-10-00267-t001:** Epidemiological characteristics of study participants (N = 293) and results of HTLV screening.

Variables	Overall (N = 293)	HTLV Positive (N = 14)	HTLV Negative (N = 279)	*p* Value
**Epidemiology**				
Sex, male, n (%)	196 (66.9%)	9 (64.3)	187 (66,9)	0.789
Age, median (IQR), years	40 (30–49)	55 (52–61)	39 (29–47)	**<0.001**
Age ≥ 50 years, n (%)	72 (24.6)	12 (85.7)	60 (21.5)	**<0.001**
**Residence, n (%)**				
Iquitos district	97 (33.1)	7 (50.0)	90 (32.3)	0.932
Punchana district	84 (28.7)	4 (28.6)	80 (28.7)	
San Juan district	64 (21.8)	1 (7.1)	63 (22.6)	
Belen district	33 (11.3)	2 (14.3)	31 (11.1)	
Outside of Iquitos city	15 (5.1)	0 (0.0)	15 (5.3)	
**Occupation, n (%)**				
Unemployed or student	111 (37.9)	5 (35.7)	106 (38.0)	0.54
Self-employment	100 (34.1)	5 (35.7)	95 (34.1)	
Cattle, agriculture or construction	47 (16.0)	3 (21.4)	44 (15.8)	
Intellectual work	28 (9.8)	0 (0.0)	28 (10.7)	
Craft work	7 (2.4)	1 (7.1)	6 (2.2)	
**Education, n (%)**				
None or only attended primary school	49 (16.7)	5 (35.7)	44 (15.8)	**0.05**
Attended secondary school or university	244 (83.3)	9 (64.3)	235 (84.2)	
**Epidemiological risk factors, n (%)**				
Breastfeeding	277 (94.5)	13 (92.9)	264 (94.6)	0.55
Blood transfusion	64 (21.8)	3 (21.4)	61 (21.9)	1.0
**Comorbidity, n (%)**				
Diabetes or high blood pressure	21 (7.2)	2 (14.3)	19 (6.8)	0.26
Digestive disease	12 (4.1)	2 (14.3)	10 (3.6)	0.10
Other cardiovascular disease	10 (3.49)	1 (7.1)	9 (3.2)	0.39
**Previous infections, n (%)**				
Strongyloides serology positive	167 (57.0)	6 (42.9)	161 (57.7)	0.29
Tuberculosis test positive	55 (18.8)	4 (28.6)	51 (18.3)	0.30
Prior gonorrhea	33 (11.3)	3 (21.4)	30 (10.8)	0.20
Prior syphilis	41 (14.0)	3 (21.3)	38 (13.6)	0.42
Chronic hepatitis	19 (6.5)	2 (14.3)	17 (6.7)	0.23
Prior cerebral toxoplasmosis	13 (4.4)	0 (0.0)	13 (4.7)	0.41
**HIV acquisition, n (%)**				
Sexual	263 (89.9)	12 (85.7)	251 (90.0)	0.71
Vertical	2 (1.0)	0 (0.0)	3 (1.1)	
Unknown	27 (9.2)	2 (14.39	25 (9.0)	
**Virology, Immunology and Adherence of Treatment**				
Nadir CD4^+^/uL, median (IQR)	228 (109–363)	213 (123–360)	230 (109–363)	0.91
Current CD4^+^, median (IQR)	446 (303–597)	455 (385–613)	441 (299–593)	0.47
Current CD4^+^ < 200/mL n (%)	22 (10.7)	0 (0.0)	22 (11.3)	0.61
Current undetectable HIV viral load (<20 copies/mL), n (%)	216 (76.3)	12 (92.39	204 (75.6)	0.31
Poor ART adherence, ≤95%), n (%)	22 (13.4)	2 (15.4)	30 (13.3)	0.89

Values with *p* value ≤ 0.05, which were subsequently included in the multivariate analysis, appear in bold. Percentages may not total 100 due to rounding.

## Data Availability

The dataset used and/or analyzed during the current study are available in the Zenodo Repository, under the ORCID: 10.5281/zenodo.14864472.

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
