# Peer review of "High Rate of Human T-Cell Lymphotropic Virus-2 in Patients with HIV in the Peruvian Amazon"

_tropicalmed, 2025, doi:10.3390/tropicalmed10090267_

Round 1
Reviewer 1 Report
Comments and Suggestions for Authors
The manuscript “High Rate of Co-Infection with Human T-Cell Lymphotropic Virus-2 in Patients with HIV in the Peruvian Amazon” addresses an important and underexplored topic: the epidemiology of HIV-HTLV coinfection in a unique geographic setting. By confirming the presence of HIV-HTLV coinfection and highlighting the predominance of HTLV-2 over HTLV-1 in this region, the study contributes valuable data to a field where information remains limited. While the study is of clear interest, the manuscript would benefit from the following refinement to improve clarity and readability:
Minor comments:
- For ELISA screening, did the authors tested sera in replicates? Please, clarify it.
- Line 89. Please, modify it from “a blood sample was obtain” to “a blood sample was obtained”.
- Data analysis: The description of age categorization is a little confusing. The percentiles (P75) are not fully explained. Please, clarify it.
- Overview of the Study Population: in the text the authors report median age of 39 years (IQR:29-47) but in Table 1 the median age is 40 years (IQR:30-49). Please, this mismatch needs correction.
- Similarly, for blood transfusion: text: 20.4% vs. table:21.8%. Please check which is correct and align them.
- The overview could highlight that this description serves as the baseline for comparing HTLV-positive vs. HTLV-negative participants preparing the reader for the subgroup comparisons that follow.
- HTLV subtypes: The text says “15 patients tested positive for HTLV for screening. Of them 14 (4.1%)”. This needs correction for clarity whether referring to prevalence among the overall screening group 14/293 (4.8%).
- Box label in the flow chart “HIV tested for HTLV” should be “293 PWH tested for HTLV”.
- Line 153. Please, correct “medium age” to “median age”.
- The discussion mentions age >50 as a risk factor, which is consistent with the results. However, age is emphasized but having no formal education or only attending primary school is not discussed.
Author Response
Please, see the attachment

Reviewer 2 Report
Comments and Suggestions for Authors
Collectively, the authors clearly addressed the study purpose and applied an appropriate study design. Moreover, they presented the results and discussion in a clear manner, comparing their findings with those of other studies, thereby providing a comprehensive picture of HTLV infection in the Peruvian Amazon. However, the following points still need to be thoroughly addressed.
Please find the full review PDFfile.

Author Response
Please, see the attachment

Reviewer 3 Report
Comments and Suggestions for Authors
The manuscript by Otero-Rodriguez et al is well written and contributes to understanding the epidemiological status of the HIV-HTLV co-infection in the Peruvian Amazon region. Nevertheless, some issues have to be addressed in order to enhance its public health impact.
Abstract: It could be improved by better emphasizing the clinical significance of the findings.
Introduction: Could benefit from a deeper discussion of the clinical implications of HTLV infection in people living with HIV.
Materials and Methods: Please, provide a sample size justification. The main concern is that testing serum on Whatman paper lacks validation, although it is mentioned as a limitation in the discussion section. The use of this method should be briefly justified (logistical constraints…). No molecular characterization is included, although it is also mentioned as a limitation. The rationale for including Strongyloides serology should be briefly explained.
Results: A multivariate logistic regression analysis should be performed to find confounders. The prevalence of HTLV needs comparison to global data. Please, double-check Table 1 (San Juan district) for numerical errors. Do the authors have data on CD8 counts ?
Discussion: The clinical impact of HTLV infection needs to be more discussed. The authors should include screening recommendations and public health implications. A deeper comparison with global data could be very informative. When discussing the "long-term non-progressor" phenotype, consider citing the paper by Abad-Fernandez et al (2022, #20) which provides mechanistic insight into how HTLV-2 might inhibit HIV progression. Also, reference #20 is misplaced in the second paragraph discussing the prevalence of HTLV-2 in the Peruvian Amazon (please, add correct references).
Author Response
Please, see the attachment
